# Effect of Land Use Land Cover and Climate Change on River Flow and Soil Loss in Didessa River Basin, South West Blue Nile, Ethiopia

**Kinati Chimdessa [1,2,*], Shoeb Quraishi [2], Asfaw Kebede [2] and Tena Alamirew [3]**

[1] Department of Natural Resources Management, Wollega University, Shambu, Ethiopia
[2] School of Water Resources & Environmental Engineering, Haramaya University, Haramaya, Ethiopia; shoeb.quraishi@gmail.com (S.Q.); asfaw649@gmail.com (A.K.)
[3] Land Resource Centre, Addis Ababa University, Addis Ababa, Ethiopia; alamirew2004@yahoo.com
[*] Correspondence: kinatichimdi@gmail.com; Tel.: +251-(0)-948589310

**Abstract:** In the Didessa river basin, which is found in Ethiopia, the human population number is increasing at an alarming rate. The conversion of forests, shrub and grasslands into cropland has increased in parallel with the population increase. The land use/land cover change (LULCC) that has been undertaken in the river basin combined with climate change may have affected the Didessa river flow and soil loss. Therefore, this study was designed to assess the impact of LULCC on the Didessa river flow and soil loss under historical and future climates. Land use/land cover (LULC) of the years 1986, 2001 and 2015 were independently combined with the historical climate to assess their individual impacts on river flow and soil loss. Further, the impact of future climates under Representative Concentration Pathways (RCP2.6, RCP4.5 and RCP8.5) scenarios on river flow and soil loss was assessed by combining the pathways with the 2015 LULC. A physically based Soil and Water Assessment Tool (SWAT2012) model in the ArcGIS10.4.1 interface was used to realize the purpose. Results of the study revealed that LULCC that occurred between 1986 and 2015 resulted in increased average sediment yield by 20.9 t ha$^{-1}$ yr$^{-1}$. Climate change under RCP2.6, RCP4.5 and RCP8.5 combined with 2015 LULC increased annual average soil losses by 31.3, 50.9 and 83.5 t ha$^{-1}$ yr$^{-1}$ compared with the 2015 LULC under historical climate data. It was also found that 13.4%, 47.1% and 87.0% of the total area may experience high soil loss under RCP2.6, RCP4.5 and RCP8.5, respectively. Annual soil losses of five top-priority sub catchments range from 62.8 to 57.7 per hectare. Nash Stuncliffe Simulation efficiency (NSE) and R$^2$ values during model calibration and validation indicated good agreement between observed and simulated values both for flow and sediment yield.

**Keywords:** Land use/Land Cover change; climate change; RCP scenarios; river flow; soil loss

---

## 1. Introduction

Hydrological processes in watersheds are affected by multitude of factors. Land use/land cover (LULC), climate, soil physico-chemical properties, geology of the land, topography, and spatial patterns of interactions among these factors are the prominent ones [1,2]. Anthropogenic interferences through land use/land cover change (LULCC) modify hydrological processes of a watershed by altering the balance between rainfall, evaporation and runoff response of an area [3,4]. This implies that knowledge of the effects of LULCC and climate change on river flow and soil loss is important for effective and sustainable land resource monitoring and planning.

Different study reports from different parts of Ethiopia indicated impacts of LULCC on river discharge. Among these isa study at the Tekeze Dam watershed, which reported an increase in mean

annual flow in the range of 129.2 to 137.7 m$^3$/s as the result of LULCC that has been undertaken between 1986 and 2008 [5]. A 12.5% increase in wet months discharge due to LULCC was reported from the Hare river watershed [6]. Another study in the Ethiopian highlands revealed 5- to 30-times higher runoff in spread agriculture than in the originally forested land [7]. Removal of forest vegetation upstream of Umbulo watershed resulted in water yield increase down-stream through the creation of a temporary water reserve, which dried up in the year 2002 [8].

Annual soil loss from the whole land mass of Ethiopia was estimated to be about 1.5 billion tons [9,10], which is equivalent to a mean annual soil loss of about 42 t ha$^{-1}$ yr$^{-1}$. A report from the Soil Conservation Research Project (SCRP) indicated that 4% of the Ethiopian highlands are so seriously eroded that they cannot be economically productive again in the foreseeable future [10]. Another study indicated that soil loss from the highlands alone was about 200 to 300 t ha$^{-1}$ yr$^{-1}$ [11]. Recent research reports from different parts of Ethiopia attributed the high rate of annual soil loss to LULCC [12–15]. On the other hand, a study by [16] attributed the high rate of soil erosion in the country to a certain combination of geomorphologic, population growth, deforestation, inappropriate cultivation practices and overgrazing.

Climate change is another equally important factor that affects catchments runoff producing certain characteristics and soil loss. Associated with climate change and variability, past findings in Ethiopia indicated historical occurrences of recurrent droughts in 11 years between the years 1952 to 2002 [17,18]. There were also occurrences of occasional floods in the years 1993, 1996, 1998 and 2006 [18], which resulted in death and resources damage. The population at risk of increased water stress in Africa due to climate change was projected to be between 75–250 million and 350–600 million by the 2020s and 2050s, respectively, [19]. Increasing temperature and variability in patterns of precipitation affect the hydrology and water resources of watersheds since they are closely linked to climate [20]. Temperature increases at the Lake Ziway watershed in Ethiopia affected hydrological processes by increasing evapotranspiration [21]. Runoff increased by over 18% for a decrease of evapotranspiration by 30% on selected catchments in the Blue Nile basin [22].

A physically based, continuous-time river basin simulation model, Soil and Water Assessment Tool (SWAT), with spatially distributed parameters operating on a daily time step [23] has been used to quantify the impact of land management practices on water, sediment and agricultural chemical yields (nutrient loss) in large and complex watersheds with varying soils, land uses and management conditions over a long period of time [24]. The model's compatibility with ArcGIS for spatial analysis, its continually available versions, embedded options for computations of hydrologic processes and its capability to partition watersheds up to the level of Hydrologic Response Units (HRU) makes the model more suitable for hydrological modeling.

The Didessa river basin contributes about one quarter of the total Blue Nile River flow as measured at the Sudan border [25] and is located above the Ethiopian Grand Renaissance Dam currently under construction on the Blue Nile River. In the Didessa district alone, which is one of the 33 districts in the river basin, the human population has increased by 40,504 between 1994 and 2007 (within 14 years' time) [26]. Forest, shrubland and grassland decreased by 11.7%, 7.1% and 7.09%, respectively, between 1986 and 2015 due to cropland expansion. Topography in the river basin is rugged, with more than 49.8% of the total river basin area with slope classes greater than 15%. Continued unsustainable resource exploitation in the river basin may result in substantial in-situ and ex-situ impacts on water and soil resources added with the current climate change. To this end, modeling LULCC and climate change impacts in the Didessa river basin is paramount. Therefore, this study was designed to assess the impacts of LULCC on Didessa river flow and soil loss under historical and future climate.

## 2. Database and Catchment Description

### 2.1. Catchment Description

The Didessa river basin is one of the fourteen sub-basins in the Blue Nile Basin. Part of the river basin addressed in this study is located between 35°56′45″ and 37°4′5″ longitude east and 7°42′2″ and 9°9′42″ latitude north (Figure 1). Its total area is estimated at 9981 km². It is characterized by a humid tropical climate with heavy annual rainfall, most of which is received during the winter rain season which extends from June to September. On the basis of altitude, the climatic condition of the sub-basin is classified as: Upland (above 2500 m a.s.l), Midland (1500 m a.s.l.–2500 m.a.s.l.) and Lowland (below 1500 m a.s.l.), which account for about 0.1%, 81.5% and 18.4% of its total area, respectively. About 46% of the total area has a slope percent greater than 15%. The mean annual rainfall of the river basin ranges between 1200 mm and 2200 mm. The maximum and minimum temperature ranges between 21.1–36.5 °C and 7.9–16.8 °C, respectively. Eastern and Western Wollega, Jimma, Illu Ababor, Buno Bedele and Kemash administrative zones are found in the riverbasin. More than 88% of the total population are rural residents and agriculture is the dominant economic activity and source of livelihood. Land use in the study river basin includes cropland, forest, shrubland, grassland, buildups and water bodies, which comprises about 56%, 19.6%, 12.4%, 6%, 5% and 0.2%, respectively, based on 2015 LULC. There are 2,992,164 total people residing in the sub-basin [26]. Geological formation in the study area is represented by the Precambrian metamorphic rocks of high and low grades with intrusive rocks, Palaeozoic sedimentary rocks, Tertiary volcanic rocks and Quaternary alluvial deposits [27].

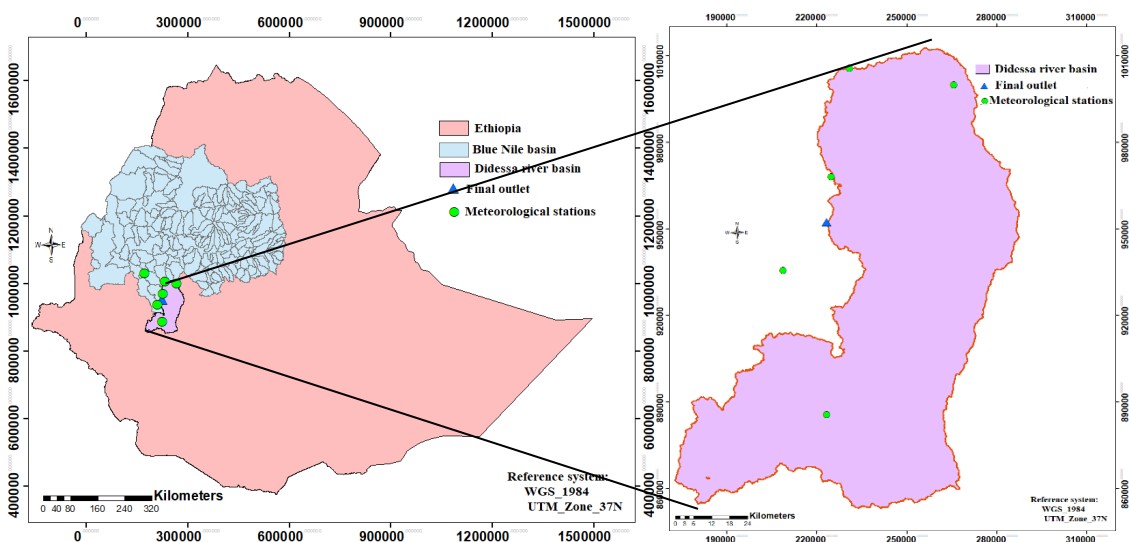

**Figure 1.** Location map of the Didessa riverbasin including the final outlet point.

### 2.2. Database

#### 2.2.1. Digital Elevation Model

A 30 m by 30 m resolution Advanced Spaceborne Thermal Emission and Reflection Radiometer (ASTER) Global Digital Elevation Model (DEM) of the Didessa river basin was downloaded from the NASA website in a raster format to calculate the flow accumulation, stream networks, and watershed delineation using SWAT watershed delineator tools. This data was projected to Universal Transverse Mercator (UTM) on spheroid of WGS84 (Figure 2).

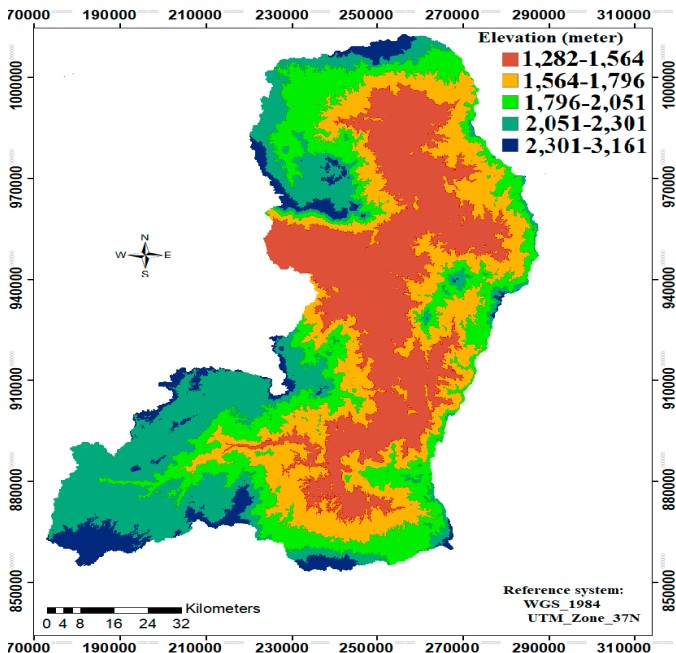

**Figure 2.** Digital elevation model of the Didessa river basin.

### 2.2.2. Climate Data

Observed data on thirty-one year's daily rainfall, maximum and minimum temperatures of the Arjo, Bedele, Dembi, Nekemte and Sire meteorological stations spanning over 1984–2014 were collected from the Ethiopian Meteorological Service Agency. Nekemte and Sire stations have complete data while Bedele has one year missing data. Arjo and Dembi have five and two years missing data, respectively. The missing historical rainfall and temperature data were filled using the first order Markov chain simulation model in Instat+v3.37 [28]. Further, the ensemble mean of daily rainfall, maximum and minimum temperature data for the period of 2018 to 2095 were collected from 17 CMIP5 multi models in MarkSim$^+$ weather file generator Google Earth interface under Representative Concentration Pathways (RCP2.6, RCP4.5 and RCP8.5) scenarios at the meteorological stations data points (Table 1). The RCPs are greenhouse gas concentration trajectory adopted by the International Panel on Climate Change (IPCC) for its Fifth Assessment Report (AR5) in 2014. The RCP2.6, RCP4.5, RCP6 and RCP8.5 imply radiative forcing of 3 W/m$^2$, 4.5 W/m$^2$, 6 W/m$^2$ and 8.5 W/m$^2$, respectively, in 2100. Lastly, both the historical and future climate data were arranged in accessible format by SWAT hydrological model for discharge and sediment simulation.

**Table 1.** List of meteorological stations and their respective data points.

| Stations | Latitude | Longitude | Altitude (m) |
|----------|----------|-----------|--------------|
| Arjo | 8°51′36″ | 36°31′12″ | 2469 |
| Bedele | 8°16′12″ | 36°12′0″ | 2030 |
| Dembi | 8°2′24″ | 36°16′12″ | 1950 |
| Nekemte | 9°13′48″ | 36°38′24″ | 2124 |
| Sire | 9°7′48″ | 36°12′36″ | 1826 |

The input annual future rainfall to the SWAT model has an increase by 10.7 mm, 12.2 mm and 54.8 mm under RCP2.6, RCP4.5 and RCP8.5, respectively, compared to the historical rainfall. The maximum temperature has an increase by 1.1 °C, 1.4 °C, and 2.3 °C under RCP2.6, RCP4.5 and RCP8.5, respectively, while the minimum temperature has decreases by 1.2 °C under RCP2.6, and increases by 0.2 °C and 1.2 °C under RCP4.5 and RCP8.5 scenarios, respectively, compared to the historical maximum and minimum temperature.

### 2.2.3. Soil Data

A soil map of the Didessa river basin with inclusive soil physical and chemical properties was extracted from Food and Agricultural Organization [29] soil in ArcGIS10.4.1 using the geo-processing program. The soil types in the study area includes Eutric fluvisols, Eutric leptosols, Eutric regosols, Eutric vertisols, Haplic acrisols, Haplic alisols, Haplic nitisols and Rhodic nitisols (Figure 3), which comprises about 0.9, 0.1, 0.02, 14, 11.6, 63.9, 6.8 and 2.5 percent, respectively, of the total area.

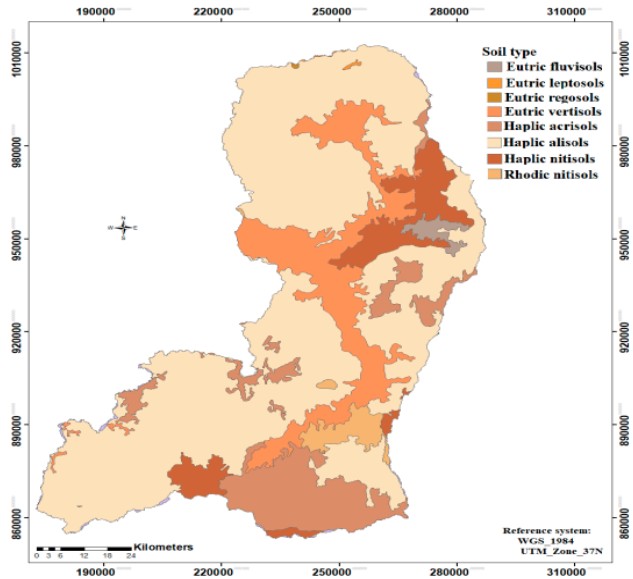

**Figure 3.** Map of major soil types in the Didessa river basin.

### 2.2.4. Land Use/Land Cover

LULC of the years 1986, 2001 and 2015 (Figure 4) assessed for their classification accuracy were inputted into the SWAT hydrological model independently to assess their effects on discharge and sediment yield under historical climates.

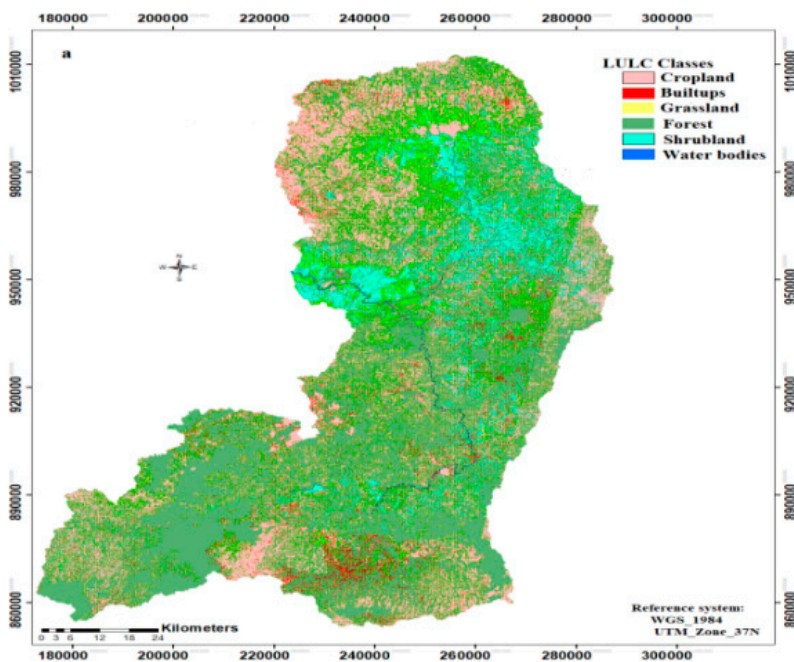

**Figure 4.** *Cont.*

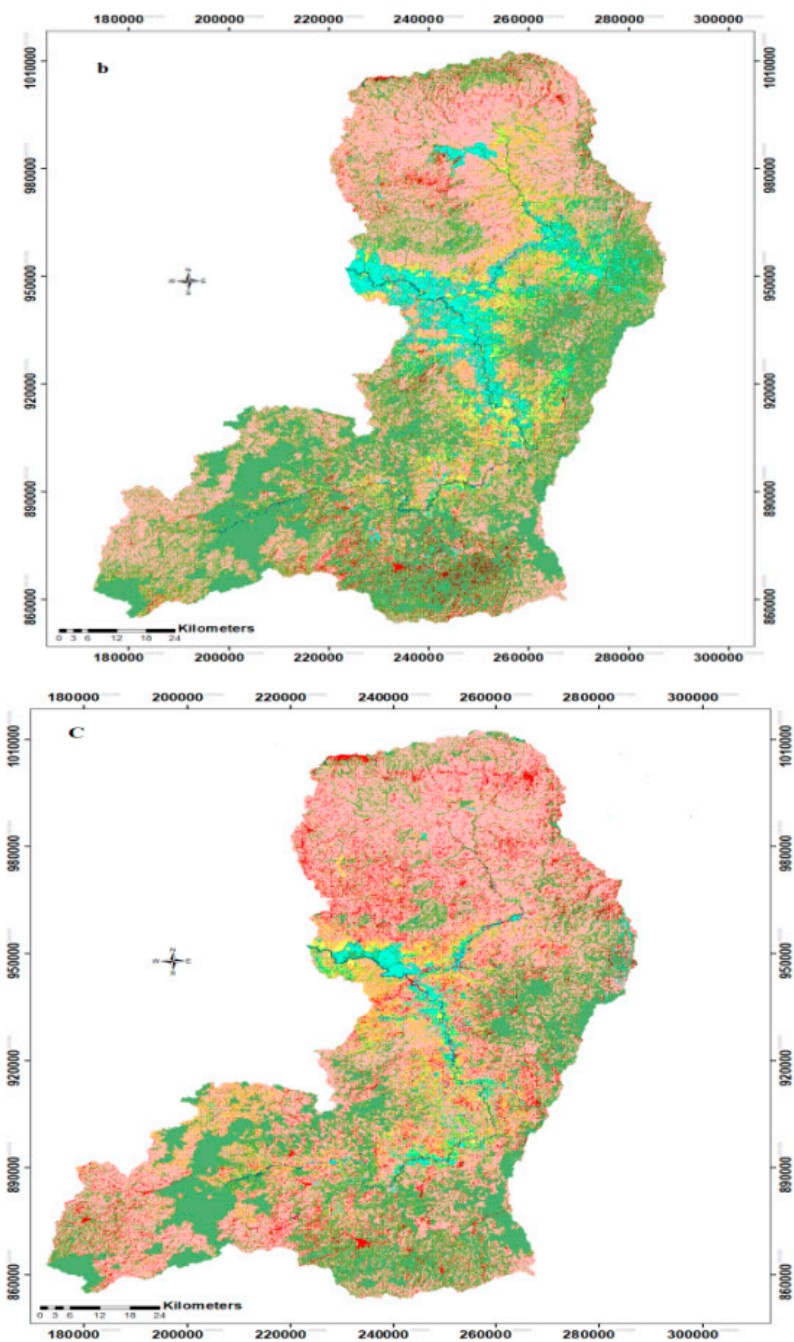

**Figure 4.** Major LULC types in the Didessa river basin in the years 1986 (**a**), 2001(**b**) and 2015 (**c**).

With regard to the types of land uses considered under each major LULC, cropland referred to land occupied by cereals, legumes and other annual crops cultivated under rainfed and irrigation systems; buildups to aerially-depicted settlements in urban and rural areas; grassland to common grazing lands in wetlands, individually owned grazing land and also savannas; forest land to natural forests, plantation forests, riverine forests, dense woodlands and also coffee forests; shrubland to dense and sparse vegetation cover of lower height in degraded areas, lowlands, and also savannas with dense mixed stands of wood and water bodies to the main Didessa River and creator lakes. The overall classification accuracy was 92.36%, 93.06% and 96.53% for the 1986, 2001 and 2015 LULC classification, respectively.

## 3. Modelling Approach

*3.1. River Flow and Sediment Yield Estimation Methods*

A physically-based Soil and Water Assessment Tool (SWAT) model developed by the United States Department of Agriculture Agricultural Research Service [30] was used to simulate the impact of LULCC and climate change on river flow and soil loss. The SWAT model simulates evapotranspiration, surface runoff, infiltration, percolation, shallow aquifer and deep aquifer flow and channel routing [23]. Runoff volume was estimated from rainfall data by using the modified Soil Conservation Service (SCS) curve number method (Equation (1)) for each Hydrologic Response Unit [31]:

$$\text{Qsurf} = \frac{(Rday - 0.2S)^2}{(Rday + 0.8S)} \tag{1}$$

where $Q_{surf}$ is the accumulated daily surface runoff (millimeters), $R_{day}$ is the daily rainfall (millimeters) and S is a retention parameter. In this method, the curve number varies non-linearly with the moisture content of the soil profile, reaching its lowest value when the soil profile moisture content approaches wilting point, and increasing to near 100 as the soil approaches saturation. The parameter S is related to curve number and was estimated by Equation (2):

$$S = \frac{25400 - 254CN}{CN} \tag{2}$$

where, CN is the curve number for the day.

The hydrologic routines, as simulated by SWAT, are based on the following water balance equation (Equation (3)):

$$\text{SWt} = \text{SW}_0 + \sum_{i=1}^{t}(\text{Rday} - \text{Qsur} - \text{Ea} - \text{Wseep} - \text{Qgw}) \tag{3}$$

where $SW_t$ is the final soil water content (mm), $SW_o$ is the initial soil water content (mm), t is time (days), $R_{day}$ is the amount of precipitation on day i (mm), $Q_{surf}$ is the amount of surface runoff on day i (mm), $E_a$ is the amount of evapotranspiration on day i (mm), $W_{seep}$ is the amount of water entering the vadose zone from the soil profile on day i (mm), and $Q_{gw}$ is the amount of return flow on day i (mm).

Sediment yield was estimated for each Hydrologic Response Unit (HRU) with the Modified Universal Soil Loss Equation (MUSLE) [32] embedded in the SWAT model (Equation (4)). In this method, sediment yield prediction was improved since runoff is a function of antecedent moisture conditions as well as rainfall energy.

$$\text{Sed} = 11.8 \times (Q_{surf} \times q_{peak} \times \text{area}_{hru})^{0.56} \times K_{USLE} \times P_{USLE} \times C_{USLE} \times LS_{USLE} \times CFRG \tag{4}$$

where Sed is sediment load (metric tons), $Q_{surf}$ is surface runoff volume (millimeter of water per hectare), $q_{peak}$ is peak runoff rate (cubic meter per second), $\text{area}_{hru}$ is HRU area (hectare), $K_{USLE}$ is soil erodibility factor, $P_{USLE}$ is support practice factor, $C_{USLE}$ is cover and management factor, $LS_{USLE}$ is topographic factor, and CFRG is the coarse fragment factor.

*3.2. Watershed Delineation*

Delineation of the watershed and sub-watersheds were done using the 30 m resolution Digital Elevation Model (DEM) in Arc SWAT following completion of SWAT project set up. There were a total of 27 sub-catchments with areas ranging between 98.71407 and 103,767.6 hectares (Figure 5).

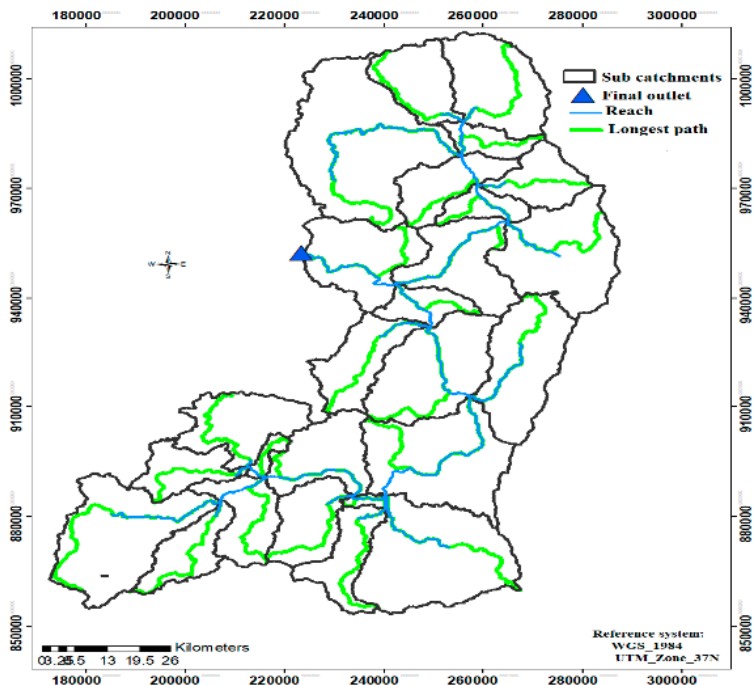

**Figure 5.** Didessa river basin with the final outlet point near Arjo town.

### 3.3. Hydrologic Response Units (HRU) Analysis

Multiple HRUs were assigned to each sub-catchment based on a 10% threshold value for LULC, soil, and slope categories as suggested by SWAT user's manual to increase chance of inclusion. The smaller proportion than the thresholds for each are shared between the major ones so that 100% of their respective areas are modeled. Accordingly, the Didessa river basin was divided into a total of 377 HRUs of different LULC, soil and slope combinations (Figure 6).

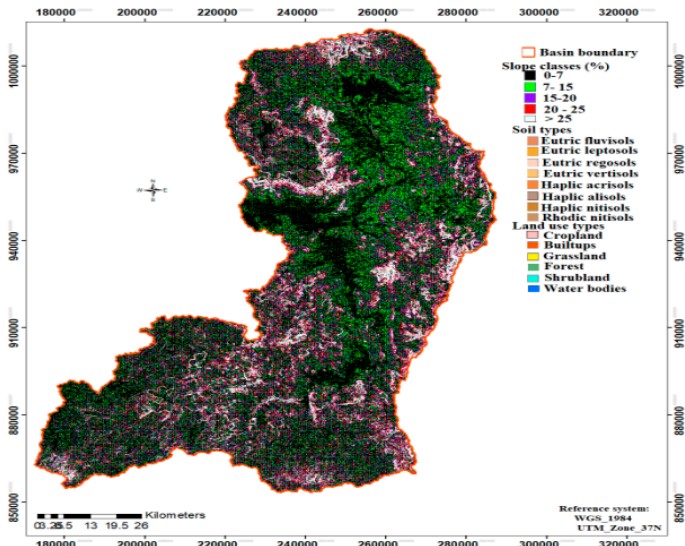

**Figure 6.** Hydrologic response units formed by overlay of slope, soil and LULC layers.

### 3.4. Parameter Sensitivity Analysis

There are a large number of SWAT model discharge and sediment parameters and it is difficult to perform calibration for all of them. Therefore, it is valuable that more determinant parameters that

affect model output are identified [33]. More sensitive parameters were identified using a sensitivity index (Equation (5)) between the allowable maximum and minimum limits for the parameters.

$$RS = \left(\frac{x}{y}\right)\left(\frac{y_1 - y_2}{x_1 - x_2}\right) \tag{5}$$

where RS is the relative sensitivity index, x is the parameter and y is the predicted output, $x_1$, $x_2$ and $y_1$, $y_2$ shows $\pm 10$ percent of the start parameters and corresponding outputparameters, respectively. RS can be small to negligible (0 < RS < 0.05), medium (0.05 < RS < 0.2), high (0.2 < RS < 1) and very high (RS > 1).

### 3.5. Model Calibration and Validation

For river flow and sediment yield simulation, a hydrological station was selected near Arjo town on the Didessa river because of data availability and a large drainage area of about 998,100 hectares behind the final outlet hydrological gauging station. Daily river flow and suspended sediment concentration were collected from the then Ethiopian Ministry of Water, Mines and Energy. Daily river flow data was converted into monthly values for model calibration and validation. With regard to sediment data, only few suspended daily average sediment concentration data collected in the years 2003, 2004 and 2005 were obtained from the Ministry office. To generate daily sediment data for the years 1992 to 2005, a regression equation (Figure 7) was established between available daily suspended sediment concentration data and corresponding daily river flow data. Monthly sediment data were derived from the generated daily sediment data. Finally, model calibration was performed using observed monthly river flow and sediment data sets ranging from 1992 to 1995, and validation using independent data sets of monthly time series ranging from 2001 to 2005. Sequential Uncertainty Fitting (SUFI-2) embedded in SWAT-CUP2012 version 5.1.6, which describes parameter uncertainty by a multivariate uniform distribution, was used during model calibration and validation [34].

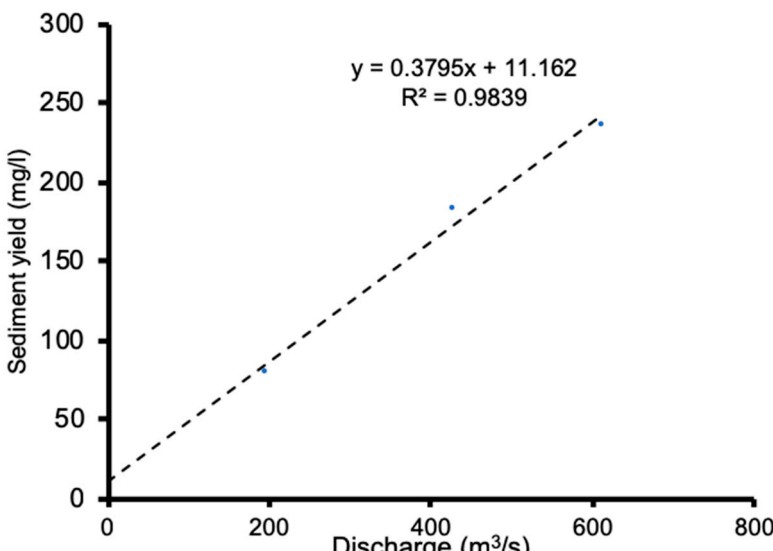

**Figure 7.** Relation between observed river flow and suspended sediment yield.

### 3.6. Model Performance Evaluation

Model performance was assessed to evaluate the model simulation outputs in relation to the observed data. The coefficient of determination ($R^2$) and the Nash and Sutcliffe simulation efficiency (ENS), $R^2$ ranges from zero to one with zero indicating poorness and one goodness of the model.

Higher values indicate less error variance values greater than 0.6 are considered acceptable [35]. $R^2$ was computed through (Equation (6)).

$$R^2 = \frac{\sum (Xi - Xav) * \sum (Yi - Yav)}{\sum \sqrt{(Xi - Xav)^2} * \sum \sqrt{(Yi - Yav)^2}} \tag{6}$$

where Xi = measured value ($m^3$/s); Xav = average measured value ($m^3$/s); Yi = simulated value ($m^3$/s) and Yav is average simulated value ($m^3$/s).

ENS indicates how well the plots of observed versus simulated data fits the 1:1 line. It was computed using Equation (7).

$$ENS = 1 - \frac{\sum (Xi - Yi)^2}{\sum (Xi - Xav)^2} \tag{7}$$

where Xi is measured value; Yi is simulated value and Xav is average observed value. The ENS value ranges from negative infinity to 1. A value of ENS less than zero depicts that the mean observed value is a better predictor than the simulated value. This indicates unacceptable performance. When the ENSvalue isgreaterthan 0.5, the simulated value is a better predictor than the mean measured value and is generally viewed as acceptable performance [35].

$$PBIAS = \frac{\sum_{i=1}^{n}(Xi - Yi) * 100}{\sum_{i=1}^{n}(xi)} \tag{8}$$

where Xi is the measured value and Yi is the simulated value.

According to [36], the optimal PBIAS value is zero, while positive and negative values indicate model underestimation and overestimation biases, respectively.

## 4. Results and Discussions

### 4.1. Identified Sensitive Parameters for River Flow

Curve number two (Cn2), soil evaporation compensation factor (Esco), soil available water capacity (Sol_awc), threshold depth of water in the shallow aquifer required for return flow (Gwqmn) and alpha base factor (Alpha_bf) were found to be more sensitivefactors with high effect on river flow. The relative sensitivity values were presented in Table 2. The highly influential parameters remained unchanged except alpha base factor, which joined the medium sensitivity class under the 2015 LULC (Table 2). However, incremental changes in the magnitude of RS was observed from 1986 LULC to 2001LULC and from 2001LULC to 2015 LULC.

**Table 2.** Relative sensitivity of SWAT parameters for river flow under land use land cover changes.

| Parameters | 1986 LULC | | | 2001 LULC | | | 2015 LULC | | |
|---|---|---|---|---|---|---|---|---|---|
| | RS | Rank | Class | RS | Rank | Class | RS | Rank | Class |
| Cn2 | 0.582 | 1 | high | 0.672 | 1 | high | 0.764 | 1 | high |
| Esco | 0.546 | 2 | high | 0.605 | 2 | high | 0.646 | 2 | high |
| Sol_awc | 0.442 | 3 | high | 0.532 | 3 | high | 0.592 | 3 | high |
| Gwqmn | 0.382 | 4 | high | 0.452 | 4 | high | 0.497 | 4 | high |
| Alpha_bf | 0.321 | 5 | high | 0.371 | 5 | high | 0.112 | 7 | med |
| Sol_z | 0.116 | 6 | medium | 0.121 | 6 | medium | 0.374 | 5 | high |
| Revapmn | 0.112 | 7 | medium | 0.025 | 10 | small | 0.035 | 10 | small |
| Gw_revap | 0.054 | 8 | medium | 0.056 | 7 | medium | 0.104 | 8 | medium |
| Hru_slp | 0.042 | 9 | small | 0.053 | 8 | medium | 0.127 | 6 | medium |
| Gw_delay | 0.014 | 10 | small | 0.032 | 9 | small | 0.053 | 9 | medium |
| Rchrg_dp | 0.012 | 11 | small | 0.013 | 12 | small | 0.013 | 12 | small |
| Surlag | 0.011 | 12 | small | 0.015 | 11 | small | 0.015 | 11 | small |

LULC = Land use/land cover; RS = relative sensitivity; CN2 = soil conservation service curve number 2; ESCO = soil evaporation compensation factor; Sol_awc = soil available water capacity;

Gwqmn = threshold depth of water in the shallow aquifer required for return flow; Alpha_bf = alpha base factor; Sol_z = soil depth; Revapmn = revaporation mean; Gw_revap= ground water revaporation coefficient; Hru_slp = hydrologic response unit slope; Gw_ delay = ground water delay; RCHRG = recharge to the deep aquifer; Surlag= surface water lag.

The highly sensitive parameters selected for final modeling of flow (Table 3) were curve number Cn2, Esco, Sol_awc, Gwqmn and Alpha_bf. These parameters were also reported as influential parameters on river discharge in the Blue Nile basin [37,38].

**Table 3.** Selected parameters for model calibration and validation.

| Parameter | 1986 LULC | | Parameter | 2001 LULC | | Parameter | 2015 LULC | |
|---|---|---|---|---|---|---|---|---|
| | Range | Fitted | | Range | Fitted | | Range | Fitted |
| Cn2 | 35–98 | 48.52 | Cn2 | 35–98 | 50.38 | Cn2 | 35–98 | 54.6 |
| Esco | 0–1 | 0.594 | Gwqmn | 0–5000 | 4454 | Esco | 0–1 | 0.74 |
| Sol_awc | 0–1 | 0.792 | Esco | 0–1 | 0.675 | Gwqmn | 0–5000 | 4465 |
| Gwqmn | 0–5000 | 4450 | Sol_awc | 0–1 | 0.836 | Sol_awc | 0–1 | 0.89 |
| Alpha_bf | 0–1 | 0.792 | Alpha_bf | 0–1 | 0.764 | Alpha_bf | 0–1 | 0.79 |

*4.2. Model Calibration and Validation for Discharge*

Hydrological models are evaluated based on their ability to capture observed data sets. Accordingly, observed and simulated discharge data have shown good agreement both during model calibration and validation (Figure 8).

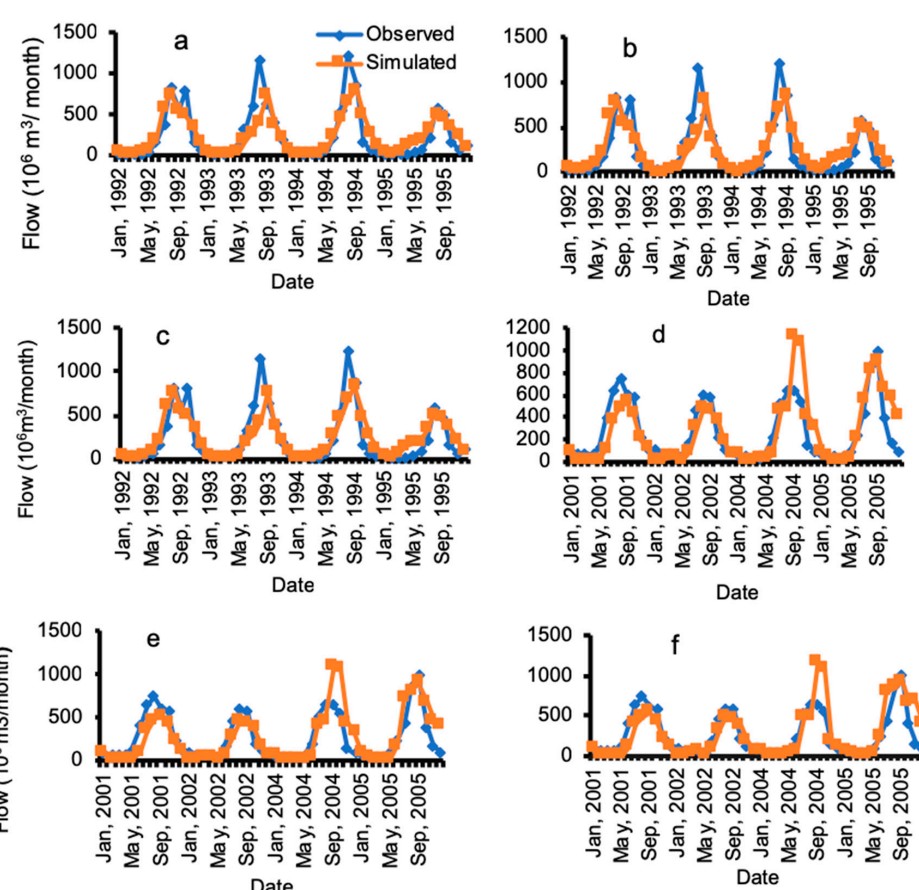

**Figure 8.** Observed and simulated flow under 1986 LULC (**a**), 2001 LULC (**b**) and 2015 LULC (**c**) during model calibration and 1986 LULC (**d**), 2001 LULC (**e**) and 2015 LULC (**f**) during model validation.

All extreme flow values correspond to observed flow during model calibration and to simulated flow during model validation under all the LULC inputs. The extreme values correspond to the month of September, which is the month when the rain season is about to cease. With respect to the simulated flow dynamics, the most extreme river flow corresponded to the 2015 LULC followed by the 2001 LULC and 1986LULC.The 2015 LULC has undergone a reduction in forest, shrub and grasslands as compared to the 1986 and 2001 LULCs.

### 4.3. Model Performance Evaluation

Comparison of observed and simulated values, both during model calibration and validation, indicated that the SWAT model can capture flow at the Didessa river basin. The agreement was reached at ENS and $R^2$ ranging between 0.5 to 0.7 and 0.8 to 0.8, respectively, both during model calibration and validation (Table 4). Previous independent study reports at the LakeTana basin and Didessa river basin in the Blue Nile basin also reported NSE and $R^2$ values ranging between 0.9 to 0.8 and 0.8 to 0.9by [37,38]. This indicates that the SWAT model can be used in planning watershed-based management interventions for better water management.

**Table 4.** Statistical values of ENS, $R^2$ and PBIAS (percent bias) during model calibration and validation under varying LULC during river discharge modeling.

| Parameter | 1986 LULC | | 2001 LULC | | 2015 LULC | |
|---|---|---|---|---|---|---|
| | Calibration | Validation | Calibration | Validation | Calibration | Validation |
| ENS | 0.7 | 0.5 | 0.7 | 0.5 | 0.7 | 0.5 |
| $R^2$ | 0.8 | 0.8 | 0.8 | 0.8 | 0.8 | 0.8 |
| PBIAS | −3.7 | −0.4 | 0.01 | −1.5 | −0.6 | −2.8 |
| Observed | 244.9 | 263.2 | 244.9 | 263.2 | 244.9 | 263.2 |
| Simulated | 236.3 | 264.4 | 241.1 | 267.3 | 246.2 | 270.5 |

### 4.4. Sensitive Parameters for Sediment Yield

A total of eight parameters (Table 5) were compared for sensitivity. Universal soil loss equation support practice factor (USLE_P), linear factor for sediment routing (Spcon), slope of the land and exponential factor for channel sediment routing (Spexp) were selected to run the model at their respective fitted values (Table 6). While USLE_P and slope are upper surface factors Spcon and Spexp are channel factors.A study from the Fincha'a watershed located in the Blue Nile basin also indicated that the Spcon and Spexp were sensitive factors besides CN and USLE_C [39].

**Table 5.** SWAT parameters and their relative sensitivity values under varying LULC for sediment yield.

| Parameters | 1986 LULC | | | 2001 LULC | | | 2015 LULC | | |
|---|---|---|---|---|---|---|---|---|---|
| | RS | Rank | Class | RS | Rank | Class | RS | Rank | Class |
| USLE_P | 0.76 | 1 | high | 0.84 | 1 | high | 0.90 | 1 | high |
| Spcon | 0.61 | 2 | high | 0.62 | 2 | high | 0.63 | 2 | high |
| Slope | 0.53 | 3 | high | 0.52 | 3 | high | 0.54 | 3 | high |
| Spexp | 0.32 | 4 | high | 0.32 | 4 | high | 0.34 | 4 | high |
| Sol_Awc | 0.14 | 5 | medium | 0.14 | 5 | medium | 0.14 | 5 | medium |
| Sol_K | 0.01 | 7 | small | 0.03 | 7 | small | 0.02 | 8 | small |
| USLE_C | 0.01 | 8 | small | 0.02 | 8 | small | 0.03 | 7 | small |
| Biomix | 0.05 | 6 | medium | 0.06 | 6 | medium | 0.05 | 6 | medium |

USLE_P = universal soil loss equation support practice factor; Spcon = linear factor for sediment routing; Slope= slope of the land; Spexp = exponential factor for sediment routing; Sol_K = soil hydraulic conductivity; USLE_C = universal soil loss equation cover factor; Biomix = bio mixture.

**Table 6.** Selected parameters fitted values during model calibration and validation for sediment yield.

| Parameters | 1986 LULC | | 2001 LULC | | 2015 LULC | |
|---|---|---|---|---|---|---|
| | Range | Fitted Value | Range | Fitted Value | Range | Fitted Value |
| USLE_P | 0–1 | 0.65 | 0–1 | 0.69 | 0–1 | 0.7 |
| Spcon | 0.0001–0.01 | 0.0074 | 0.0001–0.01 | 0.0079 | 0.0001–0.01 | 0.0081 |
| Slope | 0–1 | 1.05 | 0–1 | 1.045 | 0–1 | 1.04 |
| Spexp | 1–2 | 2.41 | 1–2 | 2.05 | 1–2 | 2.08 |

### 4.5. Model Calibration and Validation for Soil Loss

Observed and simulated sediment yield had shown good agreement both during model calibration and validation (Figure 9) with ENS and $R^2$ values ranging between 0.64 to 0.74 and 0.83 to 0.89, respectively, (Table 6). A study report from the Fincha'a watershed located in the Blue Nile Basin also showed that the SWAT model predicted sediment yield with good agreement between observed and simulated values [39]. The most extreme values were observed from the simulated sediment yield during model calibration and from observed sediment yield during model validation. The extreme values were observed during the months of August and September, in the calibration and validation periods, respectively. In the study area, August is the month where rainfall amount peaks and the soil surface is looser because of land preparation for sowing crops. September is the month when the early grown crops provide sufficient protective ground cover.

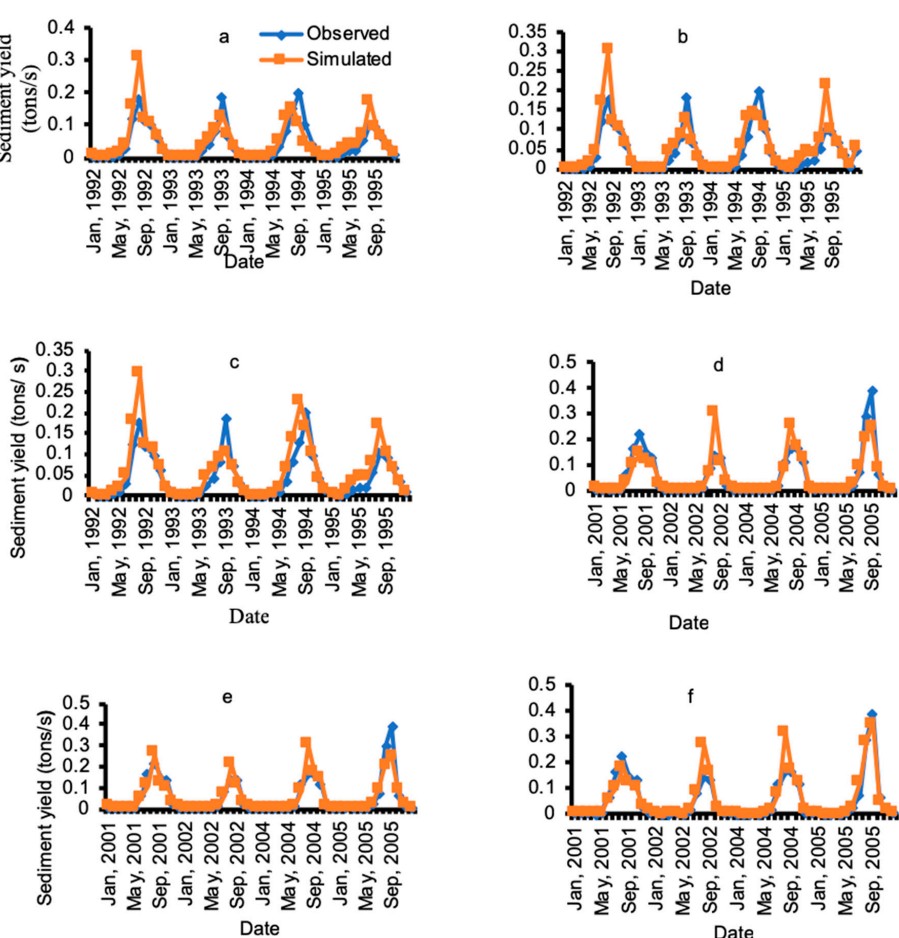

**Figure 9.** Observed versus simulated sediment yield during model calibration figures (**a–c**) and during model validation figures (**d–f**) under 1986, 2001 and 2015 LULC, respectively.

The agreement between observed and simulated sediment yield was reached at ENS and $R^2$ values ranging between 0.64 to 0.73, and 0.83 to 0.89, respectively, both during model calibration and validation (Table 7).

**Table 7.** Statistical values of ENS, $R^2$ and PBIAS during model calibration and validation under varying LULC during sediment yield modeling.

| Parameters | Calibration | | | Validation | | |
|---|---|---|---|---|---|---|
| | 1986 LULC | 2001 LULC | 2015 LULC | 1986 LULC | 2001 LULC | 2015 LULC |
| ENS | 0.68 | 0.67 | 0.6 | 0.73 | 0.71 | 0.67 |
| $R^2$ | 0.87 | 0.89 | 0.89 | 0.85 | 0.84 | 0.83 |
| PBIAS | −4.46 | −5.43 | −7.51 | −0.03 | −2.03 | −6.01 |
| Observed average | 0.044 | 0.044 | 0.044 | 0.051 | 0.051 | 0.051 |
| Simulated average | 0.05 | 0.055 | 0.06 | 0.053 | 0.056 | 0.058 |

*4.6. Added Effect of Future Climate on River Flow and Soil Loss*

Simulated average monthly river flow at the final sub-basin outlet increased by 4.9, 5.7 and 10.6 m$^3$/s between 1986 and 2001, 2001 and 2015, and also in the long-term between 1986 and 2015, respectively, because of the LULCC that occurred between these years (Table 8). The increase in river flow is asymmetrical with forest, grassland and shrubland cover, while it was found to be symmetrical with cropland and buildups cover change. A report from the Upper Fenhe river watershed which showed more increases in water yield due to afforestation than any other LULC [40] is an implication of runoff decrease, which is in agreement with this finding.

**Table 8.** Flow and soil loss at the final Didessa river outlet under LULCC and climate change.

| Model Inputs | Average Monthly Flow (m$^3$/s) | Average Sediment Yield (ton/month) |
|---|---|---|
| 1986 LULC | 263.519 | 192,370 |
| 2001 LULC | 268.406 | 193,941 |
| 2015 LULC | 274.118 | 195,466 |
| RCP2.6 (2021 to 2045) | 287.896 | 2,149,784 |
| RCP2.6 (2046 to 2070) | 281.463 | 2,138,041 |
| RCP2.6 (2071 to 2095) | 263.162 | 1,949,804 |
| RCP4.5 (2021 to 2045) | 339.217 | 2,746,660 |
| RCP4.5 (2046 to 2070) | 340.495 | 2,722,949 |
| RCP4.5 (2071 to 2095) | 316.520 | 2,475,240 |
| RCP8.5 (2021 to 2045) | 319.786 | 2,497,640 |
| RCP8.5 (2046 to 2070) | 332.0478 | 2,680,347 |
| RCP8.5 (2071 to 2095) | 347.8888 | 2,931,043 |

RCP = representative concentration pathway.

Monthly river flow increased by 3.4, 57.9 and 59.1 m$^3$/s as the result of changes in historical climate to future climates under RCP2.6, RCP4.5 and RCP8.5 scenarios, respectively (Table 7). Similar to river flow, average annual sediment production also increased by 124,546, 693,619 and 748,346 tons due to historical climate changeto futureclimate under RCP2.6, RCP4.5 and RCP8.5, respectively, when 2015 LULC is kept constant (Table 7). A study report from the Lake Ziway watershed indicated that temperature has affected hydrological processes by increasing evapotranspiration [22]. Hypothetical scenarios in the range of minus to plus thirty for both precipitation and potential evapotranspiration on selected catchments in the Blue Nile basin have also shown increment of runoff by over 18% for a decrease of evapotranspiration by 30% [23]. It is also possible to compare RCPs regarding their impact on monthly river flow and annual soil loss. Accordingly, river flow and soil loss are more enhanced under RCP8.5 compared with other two (Table 7).

Many existing reports around the world also indicated impact of climate change on water. One of these reports was an increase in projected floods by 25% to 47% between 2009 and 2099 from the United States [41]. A study from Kenya also showed an increase in stream flow parallel to rainfall and temperature increases from the 2020s to 2050s [42]. A similar study report from Pakistan reported an increase in mean annual flow under RCP 4.5 and RCP 8.5 scenarios between the years 2011–2040, 2041–2070, and 2071–2100 [43].

### 4.7. Erosion-Prone Priority Sub-Catchments

LULCC from 1986 to 2001, 2001 to 2015 and 1986 to 2015 increased average catchment soil losses by 9.6, 11, and 20.9 t ha$^{-1}$ yr$^{-1}$, respectively, (Table 9). Though not directly comparable, a study from the Tekeze Dam watershed also indicated an increase in soil loss in the range of 12.5 to 15.2 t ha$^{-1}$ yr$^{-1}$ due to LULCC [5].

**Table 9.** Effect of LULCC on catchment soil loss and treatment priority order for the nearly current 2015 LULC.

| Catchment | Area (ha) | Soil Loss (t ha$^{-1}$ yr$^{-1}$) | | | Priority Order Based on 2015 LULC |
|---|---|---|---|---|---|
| | | 1986 LULC | 2001 LULC | 2015 LULC | |
| 1 | 30,052.2 | 43.4 | 57.9 | 62.8 | 1 |
| 2 | 47,427.1 | 30.2 | 42.8 | 39.9 | 19 |
| 3 | 12,970.5 | 11.2 | 34.2 | 51.3 | 9 |
| 4 | 103,767.6 | 45.6 | 54.7 | 62.0 | 2 |
| 5 | 26,998.7 | 32.9 | 52.0 | 61.7 | 3 |
| 6 | 23,887.1 | 20.7 | 45.2 | 51.2 | 10 |
| 7 | 12,854.6 | 14.3 | 28.1 | 47.8 | 12 |
| 8 | 62,564.2 | 37.5 | 35.0 | 35.2 | 22 |
| 9 | 42,130.1 | 19.7 | 23.9 | 39.6 | 20 |
| 10 | 45,051.8 | 18.7 | 28.8 | 51.7 | 8 |
| 11 | 155,66.6 | 11.3 | 12.9 | 34.7 | 24 |
| 12 | 31,529.7 | 30.2 | 41.2 | 53.2 | 7 |
| 13 | 59,221.7 | 26.7 | 29.6 | 60.9 | 4 |
| 14 | 53,713.7 | 42.3 | 48.7 | 54.9 | 6 |
| 15 | 37,301.8 | 18.8 | 29.4 | 34.9 | 23 |
| 16 | 22,202.9 | 12.2 | 20.9 | 38.0 | 21 |
| 17 | 4773.4 | 20.3 | 32.5 | 46.1 | 14 |
| 18 | 71,506.3 | 20.3 | 37.6 | 29.3 | 26 |
| 19 | 98.7 | 9.7 | 15.7 | 41.0 | 18 |
| 20 | 35,220.8 | 28.4 | 40.8 | 44.3 | 15 |
| 21 | 2705.4 | 11.9 | 15.6 | 43.6 | 16 |
| 22 | 28,125.9 | 32.5 | 33.7 | 57.7 | 5 |
| 23 | 29,601.2 | 44.3 | 43.7 | 42.1 | 17 |
| 24 | 23,676.0 | 1.3 | 18.0 | 32.3 | 25 |
| 25 | 25,486.7 | 21.8 | 27.1 | 47.6 | 13 |
| 26 | 69,494.0 | 31.7 | 45.4 | 48.9 | 11 |
| 27 | 80,171.1 | 32.1 | 34.1 | 22.4 | 27 |
| Average | | 24.8 | 34.4 | 45.7 | |

Based on the 2015 LULC annual average per hectare soil loss, catchments 1, 4, 5, 13, 22, 14, 12, 10, 3 and 6 were identified to be the first top ten catchments, which need priority in resource allocation for treatment.

The priority order given may be subject to change if there has been effective soil and water conservation practices installed prior to 2015 in some parts of the riverbasin. Catchment numbers were provided in order to indicate the whereabouts of priority sub-catchments for treatment (Figure 10).

Annual average per hectare catchments' soil loss was further enhanced when the nearly current 2015LULC was combined with scenario-based future climates. Accordingly, average catchment soil loss

increased by 31.3, 50.9, and 83.5 t ha$^{-1}$ yr$^{-1}$ under RCP2.6, RCP4.5 and RCP8.5, respectively, compared with that under historical climate data (Table 10). Here, a symmetrically increasing relationship was observed between precipitation, temperature and soil loss. Reports indicated that global changes in temperature and precipitation patterns will impact soil erosion through multiple pathways [19].

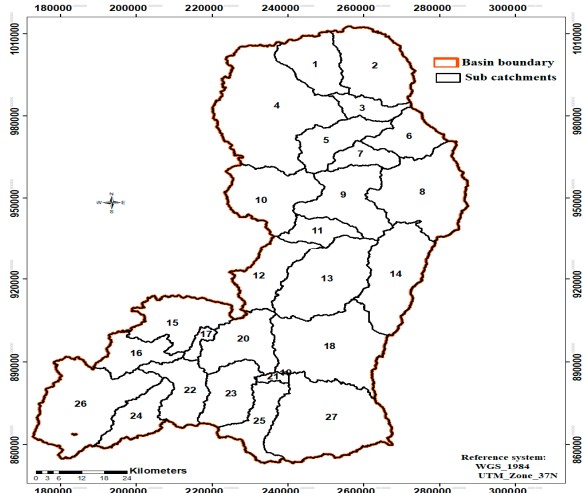

**Figure 10.** Illustrative catchment numbers to identify priority orders for treatment.

**Table 10.** Soil loss variation under constant LULC and changing climate under RCP scenarios.

| Catchment | 2015LULC & HC | | 2015 LULC & RCP2.6 | | 2015LULC & RCP4.5 | | 2015LULC & RCP8.5 | |
|---|---|---|---|---|---|---|---|---|
| | Soil Loss (t ha$^{-1}$ yr$^{-1}$) | Rank | Soil Loss (t ha$^{-1}$ yr$^{-1}$) | Rank | Soil Loss (t ha$^{-1}$ yr$^{-1}$) | Rank | Soil Loss (t ha$^{-1}$ yr$^{-1}$) | Rank |
| 1 | 62.8 | 1 | 106.5 | 1 | 134.2 | 1 | 176.1 | 1 |
| 2 | 39.9 | 19 | 83.4 | 11 | 111.3 | 6 | 157.2 | 5 |
| 3 | 51.3 | 9 | 96.7 | 5 | 124.3 | 3 | 163.9 | 4 |
| 4 | 62.0 | 2 | 102.7 | 2 | 132.4 | 2 | 178.2 | 2 |
| 5 | 61.7 | 3 | 98.9 | 3 | 126.1 | 4 | 170.1 | 3 |
| 6 | 51.2 | 10 | 86.9 | 9 | 111.1 | 8 | 151.5 | 7 |
| 7 | 47.8 | 12 | 88.2 | 7 | 109.8 | 9 | 152.2 | 6 |
| 8 | 35.2 | 22 | 70.9 | 17 | 95.3 | 14 | 131.5 | 13 |
| 9 | 39.6 | 20 | 76.4 | 14 | 99.2 | 12 | 136.6 | 12 |
| 10 | 51.7 | 8 | 89.1 | 6 | 111.2 | 7 | 143.5 | 10 |
| 11 | 34.7 | 24 | 69.8 | 19 | 89.3 | 16 | 121.9 | 15 |
| 12 | 53.2 | 7 | 87.7 | 8 | 106.0 | 10 | 141.4 | 11 |
| 13 | 60.9 | 4 | 96.9 | 4 | 116.1 | 5 | 147.0 | 8 |
| 14 | 54.9 | 6 | 83.8 | 10 | 105.4 | 11 | 144.6 | 9 |
| 15 | 34.9 | 23 | 65.3 | 24 | 84.5 | 21 | 113.5 | 19 |
| 16 | 38.0 | 21 | 70.1 | 18 | 88.0 | 17 | 117.0 | 18 |
| 17 | 46.1 | 14 | 78.6 | 13 | 97.3 | 13 | 124.8 | 14 |
| 18 | 29.3 | 26 | 58.7 | 25 | 76.7 | 25 | 106.7 | 23 |
| 19 | 41.0 | 18 | 67.3 | 21 | 86.5 | 20 | 118.7 | 17 |
| 20 | 44.3 | 15 | 72.7 | 15 | 87.2 | 18 | 112.1 | 21 |
| 21 | 43.6 | 16 | 71.3 | 16 | 86.7 | 19 | 112.3 | 20 |
| 22 | 57.7 | 5 | 79.3 | 12 | 93.9 | 15 | 121.0 | 16 |
| 23 | 42.1 | 17 | 63.3 | 23 | 80.3 | 22 | 108.4 | 22 |
| 24 | 32.3 | 25 | 49.8 | 26 | 62.4 | 26 | 85.8 | 26 |
| 25 | 47.6 | 13 | 67.1 | 22 | 77.8 | 24 | 98.4 | 25 |
| 26 | 48.9 | 11 | 67.6 | 20 | 78.5 | 23 | 101.7 | 24 |
| 27 | 22.4 | 27 | 32.1 | 27 | 39.5 | 27 | 53.15 | 27 |
| Average | 45.7 | | 77.1 | | 96.7 | | 129.2 | |

Similar to amounts of annual discharge and soil losses, soil loss (SL) severity classes also changed with LULCC and climate change. Under the 1986 LULC about 199,635.5 (21.5%) hectares of the total river basin area were classed under the none to slight (0–20 t ha$^{-1}$ yr$^{-1}$) soil loss severity class. About 728,901.8 hectares, which constitutes about 78.5% of the river basin total area, fell in the moderate

(20 to 50 t ha$^{-1}$ yr$^{-1}$) soil loss severity class. Due to LULCC between 1986 and 2001, the percentage of areas under moderate and moderately high erosion severity classes increased by 1.2% and 16.1%, respectively, while those under none to slight soil loss severity class decreased by 17.3% (Table 11). Further changes in LULC to that in 2015 resulted in total absence of areas under the none to slight severity class and decreases in those under moderate soil loss severity classes (Table 11).

**Table 11.** Percentage proportions of areas in different soil loss severity classes under LULCC and climate change based on FAO soil loss category [10].

| Soil Loss (t ha$^{-1}$ yr$^{-1}$) | Soil Loss Severity | 1986 LULC | 2001 LULC | 2015 LULC | 2015 LULC & RCP2.6 | 2015 LULC & RCP4.5 | 2015 LULC & RCP8.5 |
|---|---|---|---|---|---|---|---|
| 0–20 | None to slight | 21.5 | 4.2 | 0.0 | 0.0 | 0.0 | 0.0 |
| 20–50 | Moderate | 78.5 | 79.7 | 58.4 | 10.4 | 8.0 | 0.0 |
| 50–100 | Moderately high | 0.0 | 16.1 | 41.6 | 76.2 | 44.8 | 12.9 |
| 100–200 | High | 0.0 | 0.0 | 0.0 | 13.4 | 47.1 | 87.0 |
| >200 | Very high | 0.0 | 0.0 | 0.0 | 0.0 | 0.0 | 0.0 |

Lastly, investigation of soil loss severity classes under combined effect of 2015 LULC and future climate change under RCP2.6, RCP4.5 and RCP8.5 scenarios indicated that none of the areas fall under the none to slight soil loss severity class. About 13.4%, 47.1% and 87.0% of the total area experience high soil loss severity under RCP2.6, RCP4.5 and RCP8.5, respectively (Table 11). Comparison of soil loss severity between the RCP scenarios indicates that there is no single area in the moderate soil loss severity class under RCP8.5, while there are about 10.4% and 8.0% of the total area, which fall in this class under RCP2.6 and RCP4.5 scenarios, respectively (Table 11). This indicates that the RCP8.5 scenario is more impactful than the other two. Generally, the proportion of areas under the high soil loss severity class increased under future RCP scenarios as compared with that under historical climate data.

## 5. Conclusions

Past LULCC in the Didessa river basin has affected both river flow and soil loss in the river basin. Future climate changes under RCP scenarios enhanced river discharge and soil loss in the river basin. Average monthly river flow increased by 4.9, 5.7 and 10.6 m$^3$/s due to LULCC between 1986 and 2001, 2001 and 2015, and in the long-term between 1986 and 2015, respectively. Similarly, average catchment soil loss in the river basin increased by 9.6, 11 and 20.9 t ha$^{-1}$ yr$^{-1}$, due to LULCC between the years 1986 and 2001, 2001 and 2015 and in the long-term between 1986 and 2015, respectively. If the nearly current 2015 LULC is maintained over the coming years up to 2095, future climate change alone increases monthly river flow by 3.4, 57.9 and 59.1m$^3$/s under RCP2.6, RCP4.5, and RCP8.5 scenarios, respectively. Average annual soil loss increases by 124,546, 693,619 and 748, 346 tons due to future climate change under RCP2.6, RCP4.5 and RCP8.5, respectively. Areas under high soil loss severity class increase to about 87% of the total area under high emission scenarios (RCP8.5). Model performance assessment indicated that the SWAT model can be used to assess the impacts of LULCC and climate change on river flow and soil loss under similar environments. The continuing change in LULC and future climate may result in further increases of the Didessa river flow and soil loss. This could pose harmful effects on the projects within and outside the river basin including the new Grand Ethiopian Renaissance Dam and also on the livelihood of the community. Therefore, it is paramount that land use in the river basin be based on a policy framework to ensure sustainable land use and reduce adverse impacts. Awareness creation training on environmental values of forests is important to strengthen the valueswhich the people give to their surrounding forests.

**Author Contributions:** The corresponding author, K.C., has designed the research project, collected the necessary data and analyzed the collected data. The three co-authors listed on the cover page played key roles in advisory service during the project design, data collection and analysis, and manuscript write-up.

**Funding:** We appreciate the financial assistance from the Ethiopian Ministry of Education and Wollega University for this study without which it was impossible to happen.

**Acknowledgments:** Haramaya University is duely acknowledged for providing me suitable working environment including house with necessary facilities.

**Conflicts of Interest:** There was no any conflict of interest among the authors of this manuscript.

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
