# Peer review of "Effect of Land Use Land Cover and Climate Change on River Flow and Soil Loss in Didessa River Basin, South West Blue Nile, Ethiopia"

_hydrology, doi:10.3390/hydrology6010002_

Round 1
Reviewer 1 Report
The present article analyzed the Soil and Water Assessment Tool (SWAT2012), contained within the ArcGis10.4.1 interface to simulate river flow and soil loss in Didessa River Basin. Although the importance of this type of study is clear and the assessment of tools for this purpose may present significant advances to the scientific knowledge, the present manuscript is not ready for publication and presents problems within rationale, structure, and grammar that must be corrected before publication. For instance, in the abstract it is written that “This study aims at separation of impacts of LULC, and climate change under RCP scenarios on river flow and soil loss to identify priority areas under high soil loss severity classes” whereas in introduction the objective is stated as “assessing the impact of Land Use Land Cover and climate change on river flow and soil using Soil and Water Assessment Tool (SWAT) hydrological model”. Please notice that these objectives are not the same. The first one indicates that it is actually possible to differentiate the causes of the studied impacts on river flow and soil loss in between climate change and Land use and land cover change (and the feedback on land use and land cover change to climate change), and that this differentiation is key to be able to identify areas under high soil loss severity classes, whereas in the introduction’s objective the reader is left with the impression the manuscript aims to analyze the joint impact of changes in both climate and land use and land cover change using a very specific tool. It’s also possible to interpret the second objective as assessing the effectiveness of the said tool. These problems of precision permeate the text and make the understanding of the hypothesis and reasoning for methodology very hard. Also, some justifications of methodology and study area are presented within the Introduction but the hypothesis and rationale for it (that is the whole focus of the introduction) is not clearly stated. Moreover, there are phrases that are constructed weirdly. Especially for a non-native English speaker, these constructions make the text less fluid than is comfortable. For instance, in the title and the very beginning of the abstract, the authors used the expression “Land Use Land Cover” which is very uncommon. I’d expect to read “Land Use and Land Cover” or “Land Use/ Land Cover”. Furthermore, the way they use the expression or the acronym LULC, it is not clear if they refer simply to LULC or to LULC change (maybe referring to changes in LULC and climate, instead of LULC and climate change(s)?). The whole manuscript should be revised in this sense. Additionally, I also don’t think it is necessary to illustrate all the used data within figures and the figures that remain within the text should be reviewed. Why are there so many significant numbers (zeroes) within the coordinate system? (you may find it useful to set the Number format, within the Reference System Properties of your ArcGIS grid). Why is the data categorized the way it is? The values used in legend do not seem to have any real meaning besides being automatically generated in ArcGIS. I also highly recommend that the reference and projection system are stated within the figure’s legends (or in the figures, not just in the text) and that these figures are presented in proportion in height and width (they are distorted to better fit the page, what makes comparing them very difficult).Author Response
To my understanding, the comments by the three academic editors contribute a lot in adding scientific value to our article. I am very much happy for their constructive feedbacks. Truly speaking, I have learned a lot from their comments. Their comments help me beyond publishing this article.
1. The rationale behind the research work made clear in the beginning of the “Abstract” and end paragraph of
the “Introduction” section.
2. The disagreement between objective presented in the “abstract” and in the end paragraph of the
“Introduction” section is corrected.
3. Abstract rewritten
4. The acronym “LULC” now stands for “Land Use/ Land Cover” and the acronym “LULCC” for “Land
Use/Land Cover Change. I have defined the acronyms once in the abstract and used thereafter by the
acronyms throughout the article.
5. I have also avoided the confusion emanated from phrases “Land use land cover and climate change” by
attaching the term “change” to both independently. I was thinking to reduce redundancy, but it creates
confusion for readers.
6. I have revised the statements in the whole text where necessary.
7. I have removed some statements from the two last paragraphs which seem methodology. But, the
remaining statements in the last paragraph of the “Introduction” section are descriptions of the basic
problems in the Didessa river basin and also the objectives (after correction).
8. Now all figures adjusted. All the figures of Didessa river basin are basically of the same origin. But I agree
that there was distortion created to fit to the spaces. Now, I have corrected as much as possible.
9. Decimal number in the DEM map and in the text corrected
10. I have included the “Reference system” to all the maps at the right lower corner
Reviewer 2 Report
This is a fairly well-written article. The intent of this study is good. Some of the study results may add to the existing knowledge. However, the following comments may enhance the readability of this article:
1. The introduction section is too lengthy and includes too much literature review. In fact, some of the introduced concepts have been over-referenced (1 concept with 5 references).
2. The reference citation format may be simplified as [1, 2, 3], instead of [1, 2, & 3]. Try to use “and” instead of “&”as much as possible.
3. All the units used should be abbreviated as “t ha-1 yr-1” instead of “tons per hectare per year”.
4. All acronym names should be defined (only once) when first appear and be used thereafter.
5. The method used in this study should be restricted to the Methodology section, not mentioning it in the Introduction section.
6. The traditional degree, minute, second format is much preferred in the location (latitude, longitude) coordination.
7. Please unify the use of “Figure” and “figure”, “Table” and “table”, “km2” and “KM2”, “rainfall” and “rain fall”.
8. Is it “winter” for months of June to September in the study area?
9. The “final outlet point” is not shown in Figure 1.
10. What is the unit for the values in thelegend in Figure 2?
11. The values for latitude and longitude is very strange in Table 1.
12. It is extremely hard to see rainfall increase values of 10.68 mm in Figure 3.
13. Which map is meant for (a), (b) and (c) in Figure 5?
14. The high significant decimal number such as “98.71407” is not meaningful. Why not use just “98.7”?
15. What is the meaning of “Delineated part” in the Figure 6 caption?
16. Figure 8 has not been referred to in the text.
17. Be consistence with the numbering system for equations, not “(6)” and “6” and number all equations listed in the text.
18. Please order the arrangement of sub-figures in sequence.
19. In Tables 9 and 10, don’t use “CN” for “Catchment Number”, because CN usually denotes “Curve Number”. Besides, why not rearrange the listing according to their “Priority”?
20. “(Table 10)” in Lines 413 and 415 should be “(Table 11)” instead.
21. Too many unnecessary reference citations. Format and style for listing the references should be uniform.
22. Some of the common, general statistical analysis method may be simplified by mentioning just its name or relevant literature citation.
23. Discussions of the reasoning behind the extreme events discrepancies (Figures 9 and 10) during the calibration procedures are lacking.
Author Response
To my understanding, the comments by the three academic editors contribute a lot in adding scientific value to our article. I am very much happy for their constructive feedbacks. Truly speaking, I have learned a lot from their comments. Their comments help me beyond publishing this article.
1. Introduction section reduced by giving more emphasis to research findings in Ethiopia. There were about 53 references used in the 1st version submitted. Now reduced to 44 including 4 new references.
2. Citation format simplified and made uniform throughout the article
3. Units (tone per hectar per year) replaced by (t ha-1yr-1)
4. Acronyms defined only once and used thereafter. But, I have redefined acronyms in the foot notes of the first tables where they appear based on others recommendation.
5. Misplaced methodologies removed or taken to the appropriate section
6. Locations expressed by degree minute second system both in the text (area description) and tables
7. Acronyms in tables foot notes italicized to differ them from the major text sections
8. Differences in a word such as “Figure and figure”, “Table and table” both in the citation and captions corrected.
9. The term “winter” refers to the rain season from June to September in Ethiopia
10. The final outlet point in the maps was written as hydrological station wrongly. Now corrected.
11. The unit for elevation (figure 2) was not described. Now corrected.
12. The values of latitude and longitude changed in to degree minute second system
13. I have checked the data labels in figure 3. 10.68 mm is the real difference.
14. The labels ‘a’, ‘b’ and ‘C’ in figure 5 refer to land use land cover of the years 1986, 2001 and 2015. Now it is visible
15. Values changed to one decimal point in text. But, in tables where lower decimal places obscure differences, the higher decimal points were maintained.
16. The study was under taken in parts of Didessa sub-basin. When I say delineated part, it is to mean that the total Didessa sub basin was not covered by this study. Now I have removed it.
17. Figure 8 has referred in middle of the paragraph
18. Now I have presented all the equation numbers inside “( )”
19. Figures arranged in alphabetical sequence.
20. Now I have used the term “catchment” instead of CN. I have arranged tables 9 and 10 so that differences for a single catchment under the different factors can be compared.
21. Wrongly referred “Table 10 ” in previous Lines 413 and 415, now cited as “Table 11”
22. There were about 53 references cited in the 1st version submitted. Now reduced to 44 including 4 new references and formats for reference listing made uniform. Except differences which emanate from the cited materials (some are organizations reports, some journals give number of pages, others give starting and end page numbers, and so on)
23. Extreme value of flow and sediment yield in calibration and validation data series described.
Reviewer 3 Report
Dear authors,
I see some merits in your paper that could give it some opportunities to be published. However, you have to work on some issues that make the paper not acceptable under this current form. Check my comments about the intro, study area and figures. Add the meaning of the letters in the table captions. Check new literature recently published about SWAT in mdpi journals similar to your paper (e.g. water for Iran) or other editorials, but add new references to the discussion, to show the novelty of your model.

Author Response
To my understanding, the comments by the three academic editors contribute a lot in adding scientific value to our article. I am very much happy for their constructive feedbacks. Truly speaking, I have learned a lot from their comments. Their comments help me beyond publishing this article.
1. Now I have described the study area
2. RCP now explained as “Representative Concentration Pathway”
3. It is relevant
4. I have rewritten the abstract with modification in the beginning
5. I have introduced that I am dealing about Ethiopia
6. I have used only numbers in citations as recommended
7. I have made statements clear by adding descriptions and some removed
8. Now I have made clear that some citations I have referred are from Ethiopia
9. The term “most sever soil erosion” removed for I lack more evidence to ascertain it.
10. The term “Meteorological data” changed to climate data
11. Latitude and longitudes described in UTM now changed to degree minute and second system
12. KM2 = changed to km2
13. I missed to convert the terms above to “>” and below to “<”. I feel sorry!
14. masl changed to m.a.s.l
15. oc changed to oC
16. One decimal point used in the text and higher decimal points remained where the difference between figures will become null because of lower decimal places
17. Description of geology and land used added
18. Figures in area descriptions presented with some spacing between them
19. The term legend removed from all the figures
20. Size of scale bar and north arrow minimized
21. Grey color in grid scale bars and others inside the figures changed to black color and font size increased for better visibility
22. The term “hydrological station” in figure 1 changed to “Final outlet”
23. Legend values corrected
24. Years with missing rainfall and temperature data described.
25. Reference added to under the section climate data (Stern, )
26. RCP described as “ Representative Concentration Pathway” under climate data section
27. Figure 3 is not a result. It is just to illustrate difference in climate data used to run model
28. I have restated the statements related with figure3.
29. One decimal point used under soil data
30. Reference added under soil data
31. Figure 4 corrected
32. Figure 5 corrected. 2015 LULC put between the two. All the figures aligned center
33. Figure 6 corrected
34. I missed the recommendation under section 3.3 (reference). I feel sorry for that!
35. Legend added to figure 7.
36. Reference System added to all the figures based on the comment from reviewer 1 in the lower right corner of the maps.
37. Figure 8 adjusted as per the recommendation. I do not want to pass here that I have faced some technical shortcomings.
38. Meaning of letters in table 2 added as italicized foot notes. I did not repeat it once I did in the first table
39. I have corrected figure 9: Borders eliminated, figures arranged as recommended, only one legend used. I have used black borders but I could not use dotted l line (technical ability). Here I have also very much challenged on some of the comments like on how to use “Circle” and suggestion like “alienate” for figures 9 and 10
40. I did not add meaning of symbol (LULC) in tables foot note for I have used it repeatedly in text. Other reviewers recommended me to define acronyms only once.
41. I have described letters used in table 5
42. I did not describe letters in table 6 for I have described them in table 5.
43. Figure 10 corrected having the same challenge as in figure 9
44. Figure 11 corrected
45. Decimal place under in text below table 10 corrected
Round 2
Reviewer 1 Report
The present version of the manuscript is much improved. However, I do not think it is ready to publication yet. I've detected some points in the text in which the phrasing is confusing and others in which the affirmations seems to not be supported by the results. Moreover, the problem with spurious decimal numbers in maps persists. I maintain that there is no need to illustrate all the data used within figures, which would make the text more concise and easy to understand. Some crucial information about the RCP scenarios and land use and land cover data used are also missing. To be complete, it is necessary to fully explain the adopted climate change scenarios and premises that lead to the values using in the models, as well as the definition of each land use and land cover class, classification methodology and assessed accuracy, as well as information about spatial scale. Moreover, the manuscript would greatly benefit from a more argumentative narrative. I'm left with a series of results that are barely discussed (what they actually mean, this is a conclusion that the authors must provide and not left for the reader to think about) or discussed/compared to other results in the literature (even though some values are presented, they were not properly discussed). In this sense, I'm missing a discussion session within the manuscript.
I've registered some specific concerns over the attached pdf document.

Author Response
Dear respected editor, I have tried to include the comments as much as possible. Checked confusions created by our statements. Both spelling and grammer errors were checked and corrected to the maximum. I have made a lot of improvement on the statements in the first version of the submission. The following are some of the improvements I made through deep reading of the article end to end. I really appreciate your inputs to bring the article to this stage. I see that the comments you have been giving me are beyond this article and can help me a lot in my future carrier.
Thank you very much!!
1. Each land use described under the section 2.2.4 following the figures. Assessed accuracy of LULC classifications also described under this section
2. Climate change scenarios described in more general manner under section 2.2.2
3. Figure 3 removed thinking that only statements are enough
4. Sequence of statements improved under climate data section
5. Under section 3.5, some statements improved
6. A concluding statement added under section 4.3 at the end
7. Similar study reports described under section 4.4 to strengthen sediment sensitive parameters
8. Similar study report described under section 4.5
9. Section 4.6 improved by describing the relation observed between land use change and soil loss
10. The first paragraph under section 4.7 supported by other findings.
11. Section 4.7 paragraph 4 also improved
12. The last statement in the abstract section improved
13. Introduction line 2 “ geological make up” changed to “geology”
14. Comma added before the word “which” where I have missed it in the previous submission.
15. “that has under taken” to “that has been under taken” under “Introduction” section line 3. In the same paragraph, line 5 , the phrase “than the” changed to “than in the’
16. Introduction section paragraph 3 line 4 improved
17. Paragraph 4 in the “Introduction “ section “is projected “ changed to “was projected”
18. The word “ Temperatures” changed to “ temperature” under Introduction section, paragraph 4 line 7
19. Under Introduction section “hydrologic response unit” paragraph 5 line 7 changed to “ Hydrologic Response Unit”
20. Under catchment description, the paragraph was not indented, now improved
Line 12 in the same section, the phrase “ for residents” of the river basin “ removed for it has been indicated in the beginning of the statement.
21. The phrase ”in meter” was removed from figure 2 caption as it is already described in the legend of the figure.
22. The phrase “ in the Didessa river basin “ removed from the section 2.2.2 line 6 as it has been already described
Statement on missing data under climate section rewritten

Reviewer 2 Report
In checking all the revisions provided, I am satisfied with your efforts in addressing all my comments. However, a thorough spell-check and moderate English editing is suggested before publication.
Author Response
Dear respected editor, I have checked both spelling and grammer errors. I have made a lot of improvement on the statements in the first version of the submission. The following are some of the improvements I made through deep reading of the article end to end. I really appreciate your inputs to bring the article to this stage. I see that the comments you have been giving me was beyond this article in my future carrier.
Thank you very much!!
1. The last statement in the abstract section improved
2. Introduction line 2 “ geological make up” changed to “geology”
3. Comma added before the word “which” where I have missed it in the previous submission.
4. “that has under taken” to “that has been under taken” under “Introduction” section line 3. In the same paragraph, line 5 , the phrase “than the” changed to “than in the’
5. Introduction section paragraph 3 line 4 improved
6. Paragraph 4 in the “Introduction “ section “is projected “ changed to “was projected”
7. The word “ Temperatures” changed to “ temperature” under Introduction section, paragraph paragraph 4 line 7
8. Under Introduction section “hydrologic response unit” paragraph 5 line 7 changed to “ Hydrologic Response Unit”
9. Under catchment description, the paragraph was not indented, now improved
Line 12 in the same section, the phrase “ for residents” of the river basin “ removed for it has been indicated in the beginning of the statement.
10. The phrase ”in meter” was removed from figure 2 caption as it is already described in the legend of the figure.
11. The phrase “ in the Didessa river basin “ removed from the section 2.2.2 line 6 as it has been already described
12. Statement on missing data under climate section rewritten
13. Sequence of statements improved under climate data section
14. Under section 3.5, some statements improved
15. A concluding statement added under section 4.3 at the end
16. Section 4.6 improved by describing the relation observed between land use change and soil loss
17. The first paragraph under section 4.7 supported by other findings.
18. Section 4.7 paragraph 4 also improved
19. Conclusions also improved
Round 3
Reviewer 1 Report
The majority of my primary concerns have been addressed. My remaining concerns are:
1) Some minor grammar problems remain;
2) There is no sense in providing so many significant numbers for the coordinate system within the figures (what is the meaning of more than six zeroes in a metric system?). Besides removing these, I also recommend improving the figure resolution;
3) Please check with the editor if 'Discussion' should be a unique section within your manuscript.